# CoFeBP Micro Flowers (MFs) for Highly Efficient Hydrogen Evolution Reaction and Oxygen Evolution Reaction Electrocatalysts

**DOI:** 10.3390/nano14080698

**Published:** 2024-04-17

**Authors:** Shusen Lin, Md Ahasan Habib, Mehedi Hasan Joni, Sumiya Akter Dristy, Rutuja Mandavkar, Jae-Hun Jeong, Young-Uk Chung, Jihoon Lee

**Affiliations:** Department of Electronic Engineering, College of Electronics and Information, Kwangwoon University, Nowon-gu, Seoul 01897, Republic of Korea; lss1187112907@gmail.com (S.L.); ahasanhabibbd1971@gmail.com (M.A.H.); mehedijoni2001@gmail.com (M.H.J.); sdristy15@gmail.com (S.A.D.); rutuja.27rrm@gmail.com (R.M.)

**Keywords:** water splitting, CoFeBP, micro flower, electrocatalyst, post-annealing

## Abstract

Hydrogen is one of the most promising green energy alternatives due to its high gravimetric energy density, zero-carbon emissions, and other advantages. In this work, a CoFeBP micro-flower (MF) electrocatalyst is fabricated as an advanced water-splitting electrocatalyst by a hydrothermal approach for hydrogen production with the highly efficient hydrogen evolution reaction (HER) and oxygen evolution reaction (OER). The fabrication process of the CoFeBP MF electrocatalyst is systematically optimized by thorough investigations on various hydrothermal synthesis and post-annealing parameters. The best optimized CoFeBP MF electrode demonstrates HER/OER overpotentials of 20 mV and 219 mV at 20 mA/cm^2^. The CoFeBP MFs also exhibit a low 2-electrode (2-E) cell voltage of 1.60 V at 50 mA/cm^2^, which is comparable to the benchmark electrodes of Pt/C and RuO_2_. The CoFeBP MFs demonstrate excellent 2-E stability of over 100 h operation under harsh industrial operational conditions at 60 °C in 6 M KOH at a high current density of 1000 mA/cm^2^. The flower-like morphology can offer a largely increased electrochemical active surface area (ECSA), and systematic post-annealing can lead to improved crystallinity in CoFeBP MFs.

## 1. Introduction

Due to massive carbon emissions along with our heavy fossil fuel energy reliance, the globe is experiencing serious environmental issues and climate changes, such as unusual temperatures, super El Niño/La Niña, torrential rain, flooding, etc. It is urgently necessary to develop renewable green energy resources to address these global climate and environmental problems and also to meet future energy demands [1,2]. Hydrogen is one of the most promising green energy alternatives owing to its high gravimetric energy density, convenient usages, and zero-carbon emissions [1,3,4]. It can offer further advantages. For example, when the green hydrogen energy system is combined solar or wind power systems, the electricity supply can become more stable and optimized in terms of on-demand usage of electricity, easy storage, and convenient delivery. The demand of hydrogen has experienced substantial growth in the last 20 years with its utilization in power, transport, fuel cells, energy storage, and so forth, indicating large economic benefits [5]. Green hydrogen with zero-carbon emissions can be produced by water electrolysis, consisting of two half electrochemical reactions of hydrogen evolution reaction (HER) and oxygen evolution reaction (OER). Currently, the Pt-, Ir-, and Ru-based electrodes are the standard benchmark catalysts, but their high costs and low earth abundance restrict the large-scale adaptation of green hydrogen [3,4]. The development of highly efficient and cost-effective alternative electrocatalysts is essential at this point. 

Transition metal (TM)-based electrodes in combination with non-metallic elements can offer promising electrocatalytic alternatives due to their excellent electrochemical properties with the abundance on earth [6,7,8]. The TMs, i.e., Co, Fe, Ni, Mn, etc., with un-filled d-orbitals can promote excellent water electrolysis capability and alkaline media stability and, thus, can be suitable for water electrolysis systems [9,10,11]. Among them, cobalt (Co) is one of most widely studied TMs due to the high adsorption and desorption rates for the reaction intermediates of the water-splitting (WS) process [12]. For instance, Co-Ni-P hollow nano-bricks exhibited low HER/OER overpotentials due to the abundant mass diffusion pathways and excellent catalytic functionalities [13]. Iron (Fe) is also an abundant TM, and Fe-contained compounds demonstrate high catalytic performances toward the HER/OER [14]. For example, Fe-Ni-Co nanostructures demonstrated low HER/OER overpotentials as well as low 2-electrode (2-E) cell voltage of 1.6 V at 10 mA/cm^2^ due to the synergy between elements and high electrochemical active surface area (ECSA) [15]. Fe can demonstrates high selectivity for various hydrocarbons and oxygenates, which makes it promising for the application of electrochemical water splitting [15]. In terms of non-metallic elements, phosphorus (P) has been one of the most widely studied components in the last few decades [6,16]. The TM-P electrocatalysts can effectively alter the surface electrochemical properties and boost the WS catalytic reactions [6,16]. They also demonstrate the ability to modify the electronic structure by adjusting the oxidation states and electron density, which can further improve the efficiency in promoting both HER/OER reactions [17]. For instance, the Co_3_O_4_P nanowires/NF demonstrated superior HER efficiency and bifunctional ability in 1.0 M KOH [18]. The induced positive charged sites on the surface can enhance the adsorption of OER intermediates such as OOH* and O*, which potentially boost oxygen generation [18]. More recently, boron (B) has been gaining increased attention as another non-metallic element for TM-based electrocatalysts. TM-B-based electrocatalysts can effectively lower the kinetic energy barrier for the HER/OER and also offer superior electrochemical stability with the efficient orbital hybridization with the TMs [19,20,21,22]. 

To this end, the combination of Co, Fe, B, and P together in a high ECSA morphology can be a viable approach to develop an advanced electrocatalyst for highly efficient water electrolysis, along with their synergy and lower cost. In this work, a quaternary CoFeBP micro-flower (MF) electrocatalyst is demonstrated by a hydrothermal synthesis approach on a porous nickel form, as illustrated in Appendix A. The best optimized CoFeBP MF demonstrates comparable 3-electrode (3-E) and 2-E water electrolysis performances, as compared to the benchmark electrodes of Pt/C and RuO_2_. For example, the CoFeBP MFs demonstrate low 3-E HER/OER overpotentials of 20 and 219 mV at 20 mA/cm^2^ and low 2-E cell voltage of 1.60 V at 50 mA/cm^2^. The improved performance can be linked to the high ECSA offered by the micro-flower (MF) morphology and synergy between the elements utilized. Also, systematic synthesis optimizations and vacuum annealing treatment can effectively reduce the point defects and improve the crystallinity of MFs, further facilitating the electrocatalytic effectiveness of electrocatalysts. 

## 2. Electrode Fabrication and Structural Analysis

### 2.1. Precursor and Reaction Parameter Optimizations for CoFeBP MF

To begin with, porous Ni foam was adapted as the substrate due to excellent conductivity and porosity for CoFeBP micro-flower (MF) synthesis, and more details about the substrate and other preparation can be found in the Appendix A. Then, a CoFeBP micro flower (MF) was fabricated by a hydrothermal reaction on Ni form, followed by annealing optimization, as illustrated in Appendix A. Various hydrothermal reaction parameters were systematically optimized, and the best-optimized sample was taken for further annealing optimization. Various hydrothermal reaction parameters and precursor concentration optimization-related data are presented in Appendix A. Hydrothermal reaction duration and temperature were optimized as seen in Appendix A, and the best-optimized sample at 140 °C for 8 h was taken for further optimization. The elemental ratio of B-P was optimized, as seen in Appendix A. The Co and Fe concentration was optimized in a total molarity of 2 mM, as seen in Appendix A, where the Co_90_-Fe_10_-B_50_-P_50_ (1.8 mM Co and 0.2 mM Fe and 6 mM of B and P) demonstrated the best HER/OER performances with overpotentials of 204 and 462 mV at 200 mA/cm^2^, as summarized in Appendix A. The precursor ratio affects the atom aggregation, nucleation, and particle growth and, thus, is critical for the surface morphology, grain size, defect formation, and crystallinity [23]. An optimal elemental ratio is important as the O* and OH* absorption energy can be tuned along with different ratios [24]. The more detailed fabrication process of CoFeBP micro flowers (MFs) can be found in the Appendix A. In addition, the surface modifiers such as urea (CH_4_N_2_O) and ammonium fluoride (NH_4_F) are adapted, as seen in Appendix A. The addition of surface modifiers can be an effective approach to modify the nucleation process for high ECSA surface morphology [25,26,27,28]. For instance, the NH_4_F can effectively tailor surface morphology and can help form 3D structures. The presence of F^−^ ions (NH_4_F → NH_4_^+^ + F^−^) can readily chemisorb elements with dangling bonds and promote the nucleation process [29]. As a result, much more pronounced MF morphologies with thicker branches were commonly observed with the NH_4_F inclusion, as seen in Appendix A. The urea_90_-NH_4_F_10_ showed the best HER/OER overpotentials of 191 and 456 mV at 200 mA/cm^2^ in the set, as summarized in Appendix A. 

### 2.2. B-P Concentration Optimization for CoFeBP MF

Figure 1 shows the CoFeBP MF electrode fabrication with the B-P concentration variation. The ratio of boric acid (H_3_BO_3_) and sodium hypophosphite (H_2_NaO_2_P) was systematically controlled at a fixed total molarity of 12 mmol (mM). Other precursors were fixed at 1 mM Co(NO_3_)_2_•6H_2_O for Co and 1 mM Fe(NO_3_)_3_•9H_2_O for Fe based on the previous parameter optimizations. Along with the B-P concentration control, various surface morphologies of CoFeBP structures were observed, as seen in Figure 1a–e. Generally, the ternary CoFeB or CoFeP showed cluster morphologies, as seen in Figure 1a,e. Then, the quaternary CoFeBP electrodes showed 3D micro-flower (MF)-like morphologies, as seen in Figure 1b–d. The B_50_-P_50_ sample demonstrated the most well-defined MF morphology with the high-density branch structures along with the well-balanced elemental ratio of B and P, as seen in Figure 1c [30]. Due to the existence of five valence electrons in the outer shell of phosphorus (3s^2^3p^3^) [31] and high affinity of boron [32], strong ionic bonding with the metallic Co and Fe can be achieved through orbital hybridization with the well-balanced B and P. Additional SEM images can be found in Appendix A. The EDS phase maps of CoFeB_50_P_50_ MFs confirmed the uniform distributions of Co Lα1, Fe Lα1, B Kα, P Kα1, and Ni Lα1, as seen in Figure 1f–f-5. Also, the EDS spectrum of B_50_-P_50_ confirmed the presence of Co Lα1, Fe Lα1, B Kα, P Kα1, and Ni Lα1 peaks in Figure 1g. EDS spectra of other samples can be found in Appendix A. Atomic (At.) % summary in Figure 1h confirms the gradual increase in P incorporation along with the increased P (0–12 mM), while the Co, Fe, and B showed gradually decreasing trends. For the CoFeB_50_P_50_ MF, 25.79 and 30.82% of B and P atoms were incorporated and 29.29 and 10.57% of Co-Fe atoms were observed.

The X-ray diffraction (XRD) of CoFeB, CoFeP, CoBP, and CoFeBP electrodes is shown in Figure 1i, and the individual XRD patterns of CoFeB, CoFeP, CoBP, and CoFeBP are shown in Appendix A. The XRD PDF cards of related compounds are shown in Appendix A. Two strong diffraction peaks were commonly observed at ~43 and 52° in all XRD patterns, which can be indexed to the (111) and (200) planes of the Ni foam substrate [33]. The polycrystalline phase was indicated with multiple diffraction peaks that did not show a match with the relevant combinations, such as Co-B, Co-P, Fe-B, Fe-P, and B-P. In detail, CoFeP exhibited strong peaks at 27.7 and 28.2°, and CoFeB possessed several intense peaks at 33.2, 35.5, and 38.7° and other secondary peaks. CoBP demonstrated stronger diffraction peaks at 27.9, 32.2, and 37.2° and other smaller peaks, which might indicate that CoBP is in a short-range polycrystalline phase [34]. The CoFeBP electrode showed slightly lower intensity and peaks at 27.0, 27.9, 34.0, and 35.2°, which also indicate a polycrystalline phase [35]. Among the 2θ range from 20 to 65°, the appearance of multiple peaks in all samples might indicate the fabrication of polycrystalline structures [36]. Notably, recent studies suggested enhanced intrinsic water-splitting (WS) activity with the polycrystalline phase, as compared to the single-crystal structures due to the increased exposure of active sites and structural stability [22,35,37]. The long-range disordered polycrystalline surface can demonstrate higher structural flexibility and stability for the water-splitting reactions [22,35,37]. A more detailed discussion of the XRD analysis can be found in Appendix A.

### 2.3. Structural Analysis of CoFeBP MF along with Post-Annealing Optimization

Figure 2 shows the CoFeBP MFs along with the post-annealing temperature optimization on the best CoFeBP MFs after the synthesis parameter optimizations. The best CoFeBP MFs, i.e., Co_90_-Fe_10_-B_50_-P_50_ electrode, were taken for annealing in a rapid thermal processing (RTP) system, and a series of annealing experiments was conducted to establish the optimal conditions, as seen in Appendix A. The optimal annealing duration was 30 min based on the optimization study between 10 and 60 min, as seen in Appendix A. Along with the post-annealing temperature variation, there was no obvious change in the surface morphology below 200 °C, as clearly seen in Figure 2a,b, and the MF morphology was well maintained. At 300 °C, the branch structure began to fall off from the MF matrix, as shown in Figure 2c. At 500 °C, severe structural damage were observed, i.e., branches were cracked due to the excessive thermal energy, as seen in Figure 2d [38]. Often, the electrocatalysts synthesized by the hydrothermal reaction can be in the polycrystalline phase [30], and the degree of crystallinity can be improved to a certain extent by moderate post-annealing treatment through adatom diffusion [30]. The charge transfer process can be promoted along with improved local crystallinity [38]. Annealing can improve the local crystallinity due to the removal of defects [39]; however, a longer duration or excessive temperature can exhibit structural damage [39]. The EDS phase maps of 100 °C MF show the uniform distribution of Co Lα1, Fe Lα1, B Kα, and P Kα1 phases in Figure 2e–e-4, and the EDS line profiles also confirmed the uniform distributions of Co, Fe, B, and P after the thermal treatment in Figure 2f-1–f-4. Additional SEM images can be found in Appendix A, and the EDS spectra for the annealing temperature variation set and summary plot of atomic percentage are provided in Appendix A. There was no obvious elemental variation in the atomic percentage, indicating no dissociation and sublimation of elements. 

Raman analysis was conducted to probe the crystal quality along with annealing, as seen in Figure 3a. The 100 °C annealing demonstrated the strongest Raman bands with Raman peaks at 261, 360, 478, 596, and 962 cm^−1^, indicating the best crystal quality of CoFeBP MFs in this set [40]. The stretching vibration peaks at 950–1100 cm^−1^ may be associated with the oxygen bonds of metallic atoms (M-O-O) as indicated with the red dashed lines in Figure 3a [41,42]. At 100 °C, the peak split into two shoulder peaks, which could be related to the symmetric stretching O-O vibration [41,42]. More detailed Raman analysis can be found in the Appendix A. Figure 3b shows the XRD spectrum of the CoFeBP MF (100 °C) electrode. After annealing, the peak intensity was clearly intensified, suggesting improved local crystallinity [40]. Also, new XRD peaks appeared, i.e., at 22.8°, 60.8°, and 66.5°, again suggesting improved crystal quality with local lattice reconstruction due to the thermal-triggered adatom diffusion [38]. The multiple peaks clearly indicate a short-range polycrystal phase of CoFeBP MF, even after annealing [38]. The X-ray photoelectron spectroscopy (XPS) was conducted to probe the chemical states of Co, Fe, B, P, and O 1s and C 1s. The full-scan spectrum is shown in Figure 3c, and the zoom-in spectra are presented in Figure 3c-1–c-4. First, the O 1s and C 1s are due to surface oxidation and reference calibration. The O 1s spectrum in Appendix A shows three distinct peaks located at 532.1, 539.9, and 534.5 eV, corresponding to M-O, M-OH, and absorbed H_2_O, respectively (M=Co and Fe) [43]. The surface oxidation is inevitable due to air exposure, and C can always be observed along with reference calibration. In the Co 2p spectrum in Figure 3c-1, the binding energy (BE) of Co 2p_1/2_ and Co 2p_3/2_ was found at 793.6 and 778.5 eV, as compared with the pristine peak obtained in the XPS handbook (793.3 and 778.3 eV) [44], exhibiting positive shifts of 0.3 and 0.2 eV. The positive shift indicates the electron donation from Co atoms [45]. In the compound molecule formation, a positive shift of elemental electronic states indicates the donation of electrons in the ionic bonding. The peaks at 789.6 and 804.7 can be indexed to the satellite peaks [45]. The obtained dominant peaks along with their satellite peaks indicate the formation of Co_3_O_4_ on the surface [45,46]. The peaks centered at 795.9 and 780.8 eV indicate the existence of CoO on the surface [45,46]. In the Fe 2p spectrum in Figure 3c-2, the peaks located at 707.3 and 720.9 eV can be assigned to Fe^0^ (Fe 2p3/2 and Fe 2p1/2). Similarly, positive shifts of 0.8 eV and 0.3 eV were observed, as compared with their pristine 720.1 and 707.0 eV, indicating electronic interaction with the non-metallic atoms [47]. The existence of 708.6 and 723.6 eV peaks represents the FeOOH, and the BEs at 712.3 and 727.2 eV can indicate the formation of Fe_3_O_4_ [48]. The peaks centered at 715.1 and 731.8 eV can be assigned to the shakeup satellite peak [49,50]. On the other hand, the B 1s level was negatively shifted by 2.4 eV (from 189.4 to 187.0 eV), indicating ionic bonding by electron acceptance. The 192.5 eV peak can be assigned to the B_2_O_3_ oxidation states [51]. In the P 2p spectrum, two dominant peaks appeared at 128.7 and 129.5 eV, which are assigned to P 2P_3/2_ and P 2P_1/2,_ respectively. The BEs of P 2P_3/2_ and P 2P_1/2_ were also negatively shifted to lower BE from the standard 129.9 to 128.7 eV and from 130.74 to 129.5 eV, implying electron acceptance to the p-orbital [52]. The strong electronegativity and electron affinity of P atoms can be beneficial for the strong bond formation [52]. The 133.4 eV peak belongs to the oxidized P species (P-O) [52]. Overall, the positive and negative shifts of elemental electronic states can imply strong ionic bonding, and oxidized states were found alongside them. More detailed XPS analysis can be seen in the Appendix A. In summary, from the structural analysis, the obtained structure should be a cobalt–iron–boron–phosphide (CoFeBP) alloy in the polycrystalline phase based on the XRD, XPS, and Raman analyses.

## 3. Electrochemical Analysis

### 3.1. The 3-E Electrochemical Properties of CoFeBP MFs

Figure 4 shows the 3-electrode (3-E) electrochemical characterizations of the post-annealing temperature variation set in 1 M KOH. The HER/OER LSV results are shown in Figure 4a–b, where the 100 °C CoFeBP MF exhibited the best catalytic performances with the lowest overpotential values of 145 and 424 mV at 200 mA/cm^2^, as summarized in Figure 4c. Figure 4d,e show the HER/OER Tafel slope values, derived based on the relation [53] ƞ = a + b log|−j|, where ƞ presents the overpotential, a is the Tafel constant, and b represents the Tafel slope. The 100 °C CoFeBP MFs showed the lowest HER/OER Tafel values of 62.0 and 199 mV/dec, as seen in Figure 4f, indicating the best charge transfer kinetics and reaction speed in this set [54]. While the HER Tafel value was quite good, the OER Tafel value was relatively high, which can negatively affect the catalytic performance in bifunctional 2-E performance. HER/OER Nyquist plots by the electrochemical impedance spectroscopy (EIS) are shown in Figure 4g,h, which can provide insights into the impedance of electrochemical systems, such as the solution resistance (Rs) and charge transfer resistance (R*_ct_*) [55]. Further, 100 °C CoFeBP MFs exhibited the lowest HER/OER charge transfer resistance (R*_ct_*) values of 7.1 and 13.3 Ω, suggesting the highest HER/OER charge transport characteristics in this set [56,57]. The HER/OER double-layer capacitance (C*_dl_*) is summarized in Figure 4i, which was derived from the CV plots via the following relation, J = (J_a_ − J_c_)/2, where J_a_ is the anodic current and J_c_ is the cathode current [57]. The corresponding CV plots are provided in Appendix A. The 100 °C MFs demonstrated the highest HER/OER C*_dl_* values of 4.6 and 14.5 mF/cm^2^, suggesting the largest electrochemical surface area (ECSA) of 100 °C CoFeBP MFs. The C*_dl_* values are specifically related to the capacitance at the electrode–electrolyte interface and, generally, the OER C*_dl_* values are higher. The HER is a 2-electron transfer process, and OER is a 4-electron transfer process. Thus, the OER requires higher voltages. This can indicate that more metallic sites participate in the OER process. With a higher voltage application, we can observe larger current changes, which are reflected as higher C*_dl_* values. The 3-E electrochemical properties of the best CoFeBP MF in 1 M KOH are summarized in Table 1.

The HER operation can be summarized: Volmer step: H_2_O + e^−^ + M → OH^−^ + MH_ad_; Heyrovsky step: H_2_O + e^−^ + MH_ad_ → M + OH^−^ + H_2_; or Tafel reaction: 2MH_ad_ → 2M + H_2_ [58]. The M is the metallic site (Co and Fe), and H_ad_ is the absorbed hydrogen proton. The OER process can be summarized: OH^−^ + M = M-OH + e^−^, M-OH + OH^−^ = M-O +H_2_O + e^−^, 2M-O → 2M + O_2_. Alternativley, M-O + OH^−^ = M-OOH + e^−^; M-OOH + OH^−^ → O_2_ + H_2_O + e^−^ + M [40,58]. It is important to have a lower energy barrier to break the covalent H-O-H bonds. The efficient H_ad_ and hydroxyl adsorption/release steps are imperative for the efficient HER/OER processes in alkaline solutions. The multi-metallic active centers in the CoFeBP MFs can effectively break the tetrahedral covalent H_2_O bond and form the metal hydride and hydroxyl intermediates (MH_ad_ and M-OOH), allowing for continuous hydrogen and oxygen generation via the HER and OER steps [59,60]. The micro-flower morphology can provides a largely increased number of absorption sites, in which the MH_ad_ and M-OOH intermediates can easily be increased [60]. The incorporation of B atoms can result in the modulation of electronic states for active sites, leading to efficient catalytic reactions and higher electronic conductivity for higher HER/OER rates [61,62]. At the same time, P is a well-known element for superior HER/OER activities with its strong electronegativity, and the inclusion of P can significantly improve the HER/OER kinetics and performances in the MF matrix [62]. 

The turnover frequency (TOF) of CoFeBP MFs is calculated ass seen in Figure 4j,k. The TOF can be utilized to reflect the intrinsic activity of electrodes along with the number of H_2_ and O_2_ molecules generated per site and time at the turnover [63]. The 100 °C electrode yields the highest HER/OER TOF values of 0.193 and 0.101 site^−1^s^−1^, as summarized in Figure 4j,k. The comparison of HER/OER TOF values with other electrocatalysts is summarized in Appendix A. The HER and OER Faradaic efficiency was calculated to assess the energy efficiency in Figure 4l [64]. The generated H_2_ or O_2_ was collected by the water displacement approach, as seen in Appendix A. The HER/OER Faradaic efficiency of optimized CoFeBP MFs was found to have relatively high efficiencies of 90.16% and 90.20% for 30 min, as summarized in Figure 4l [14]. The efficiency loss could be due to the heat generation and bubble formation [65]. More details on the TOF and Faradaic efficiency calculations can be found in the Appendix A. 

In addition, the formation of oxidation peaks was observed in the OER LSV curves, as seen in Figure 4b. The oxidation peak gradually increased along with the increased performance of samples, and a higher oxidation current was observed with a better-performing sample. To observe the oxidation and reduction peak formation in the same range of OER, cyclic voltammetry (CV) was performed, as seen in Appendix A. The oxidation state was observed at ~1.48 V, which can be due to the formation of M-OOH (Co and Fe), i.e., cobalt hydroxide (CoOOH), cobalt oxide (Co_3_O_4_), iron hydroxide (Fe-OOH), and other Fe-oxide species [66]. During the positive scan, the formation of a 1.48 V peak can be due to the formation of M-OOH, i.e., (M(OH)_2_ + OH^−^ → MOOH + H_2_O + e^−^; 3M(OH)_2_ + 2OH^−^ → M_3_O_4_ + 4H_2_O + 2e^−^) [67,68]. In the process of a negative scan, reversible reactions can take place with the reduction peak at ~1.39 V, and the Fe^−^ and Co-oxide species can be reduced to M(OH)_2_. The Fe/Co oxide peak formation can improve the OER performance as the M-OOH (Co, Fe) can directly take part in the OER process as an OER intermediate and can function as active OER centers, further boosting the splitting process [67,68].

Overall, the 100 °C CoFeBP MFs demonstrated the best electrochemical activities in this set, which are mainly due to the improved crystallinity and effectively boosted carrier transport in moderate annealing conditions [69]. Specifically, during the vacuum annealing process, hydroxyl groups can be eliminated, and the oxygen vacancies (O_vac_) can be removed [54,70]. The defect density can be altered by atomic diffusion, and thermal treatment is crucial in the modulation of surface active sites [40]. Nevertheless, excessive thermal energy can damage MF structures, indicating limited performance [40]. XRD, Raman, EIS, and LSV analyses before/after annealing on CoFeBP MFs can be found in the Appendix A. 

### 3.2. The 3-E LSV Activity of CoFeBP MFs in Different pH Media

Figure 5 shows the 3-E HER/OER performances of the best CoFeBP MF in different pH electrolytes, as compared with the Pt/C and RuO_2_ benchmark electrodes. The alkaline, acidic and neutral media were prepared as 1.0 M KOH, 0.5 M H_2_SO_4_, and 1.0 M PBS solutions. The reference electrode fabrication of Pt/C and RuO_2_ can be found in the Appendix A. As seen in Figure 5a–f, the 3-E HER/OER performances were the best in alkaline conditions and worst in the neutral medium in the order of 1 M KOH > 0.5 M H_2_SO_4_ > 1 M PBS. The reference electrodes of Pt/C and RuO_2_ demonstrated better HER/OER performances in all electrolytes due to the excellent inherent electrocatalytic properties of reference electrodes. Meanwhile, the CoFeBP MFs demonstrated quite comparable HER/OER performances in the alkaline media in Figure 5a,d: 196 and 130 mV for HER and 397 and 491 mV at 300 mA/cm^2^. The lower performance in the acidic media could be due to the high concentrations of H^+^ ions, and, generally, performance degradation and corrosion of transition metal-based electrodes can be observed in acidic solutions [71]. The neutral media showed the worst performance due to the lack of conductive ions [40]. The 3-E steady-state measurements in alkaline media (1.0 M KOH) were carried out further to understand the stability of CoFeBP MFs at various overpotentials (Figure 5g). The CA measurement checks the initial fluctuation and compares the result to LSV. The 3-E CA response demonstrated a negligible difference with LSV, as summarized in Appendix A, confirming the excellent early-stage stabile operation of CoFeBP MFs. The HER/OER performance of CoFeBP MFs is compared with the state-of-the-art transition metal-based electrodes at 20 mA/cm^2^ in Figure 5h,i and Table 2 and Table 3. The CoFeBP MF demonstrated an excellent HER performance of 20 mV at 20 mA/cm^2^, which ranked it as the second-best HER electrode as compared with the state of the art in Figure 5h and Table 2. At the same time, CoFeBP MF demonstrated a decent OER performance of 219 mV at 20 mA/cm^2^, which ranked it as the sixth-best transition metal-based OER electrode (Figure 5i and Table 3). However, the OER performance was not as high as HER. 

### 3.3. The 2-E Activity of CoFeBP MFs

Figure 6a–c show the 2-E LSV performances of CoFeBP MFs in different pH electrolytes, as compared with the benchmark electrodes. Both systems, i.e., CoFeBP‖CoFeBP and Pt/C‖RuO_2_, demonstrated a 2-E performance in the order of 1 M KOH > 0.5 M H_2_SO_4_ > 1 M PBS. At the same time, CoFeBP‖CoFeBP demonstrated comparable performances to the benchmarks in all pH media. The 2-E CA response in 1 M KOH is provided in Appendix A, and stable output can be maintained at various cell voltages, again confirming the excellent stability of CoFeBP MFs. The 2-E LSV measurement was extended up to 1000 mA/cm^2^ and 6 M KOH LSV at 60 °C, as shown in Figure 6d. A high-current performance (>1000 mA/cm^2^) is imperative for industrial applications with a high gas generation rate [1]. Here, the CoFeBP‖CoFeBP MFs demonstrated quite a comparative performance with the Pt/C‖RuO_2_ with the 2-E cell voltage of 2.66 V at 1000 mA/cm^2^ in 1.0 M KOH. The CoFeBP‖CoFeBP MFs exhibited a lower 2-E cell voltage of 2.54 V at 1000 mA/cm^2^ in 6 M KOH at 60 °C, which compares to the 2.43 V of benchmarks. The better LSV performance can be due to the improved reaction kinetics by the high solution conductivity with high OH^−^ concentrations and high temperature [40]. The water electrolysis industry utilizes strong alkaline media and higher operational temperature due to the better water-splitting (WS) performance [64,79]. Further, 100 h stability operation was carried out in the industrial operational conditions in 6 M KOH at 60 °C in Figure 6e. The 100 h stability operation of 2-E CoFeBP‖CoFeBP MFs clearly confirmed the superior stability of CoFeBP MFs with the very stable current, with only a minor current fluctuation at 1000 mA/cm^2^ in the industrial operational conditions. There was no significant dissolution of CoFeBP MFBs under the industrial operational condition. After the continuous 100 h operation, small particles were observed at the bottom of the cell. The CoFeBP MFs demonstrated excellent repeatability, with minor differences after 1000 cycles in Figure 6f, further confirming the excellent stability of CoFeBP MFs. In addition, the 12 h stability tests in 1 and 6 M KOH at 1000 mA/cm^2^ at room temperature also confirmed excellent stability, as seen in Appendix A. 

The water splitting (WS) of CoFeBP MFs in natural seawater (SW) and river water (RW) with the addition of 1.0 M KOH was performed, as seen in Figure 6g,h. Bare SW and RW performances can be seen in Appendix A. Generally, CoFeBP MFs demonstrated higher WS performances in SW over RW. The 1.0 M KOH addition largely improved the WS performance due to the boosted solution conductivity by the increased OH^−^ ions. The 2-E cell voltages of 3.87 and 2.34 V in RW and SW were obtained at 200 mA/cm^2^ by the CoFeBP MFs, as seen in Figure 6g,h. The higher WS performance in SW can be due to the presence of conductive ions, such as Na^+^ and Cl^−^ [40]. In natural waters, a lot of ions and biological compounds and elements, such as Mg^+^, F^−^, Ca^+^, Br^−^, bacteria, and dust particles exist, and, thus, the WS performance in natural waters is generally much worse than DI water-based solutions [40]. The CA response in SW + 1.0 M KOH was measured, as seen in Appendix A. Stable current output was obtained in 30 min continuous operations at various voltages, which confirmed the stable operation of CoFeBP MFs in natural waters. Overall, the CoFeBP MF showed WS performance and stability in real waters. The 2-E cell voltages in various electrolytes and natural waters are summarized in Table 4. Finally, the CoFeBP MF demonstrated an outstanding 2-E cell voltage of 1.60 V at 50 mA/cm^2^, and CoFeBP MF may be one of the best transition metal-based electrocatalysts, as compared with the state of the art, as summarized in Figure 6i and Table 5. 

### 3.4. Characterization after Stability Test

The CoFeBP MFs (anode) are characterized before/after 12 h stability operation in 1 M KOH at 1000 mA/cm^2^ in Figure 7. Anode data are shown here as more obvious changes can be generally observed on the anode side. Additional data can be seen in Appendix A. Firstly, the SEM images showed no visible difference before/after the long stability operation in Figure 7a,a-1. Upon closer viewing, the edges of branches became slightly rougher after the 12 h stability operation; however, the micro-flower structures were well maintained. This indicates the good structural stability of CoFeBP MFs. The SEM images of the cathode and anode after the 12 h stability operation are shown in Appendix A. There was also no obvious difference observed in either electrode. The Raman spectrum of the CoFeBP MF anode after stability testing is shown in Figure 7b. Raman bands of the anode and cathode after the 12 h stability test are compared in Appendix A. The Raman of CoFeBP MF before the stability test can be seen in Figure 3a. In general, the Raman intensity was largely reduced after the 12 h stability test. The Raman intensity reduction could be due to the formation of oxide species after the long-term stability operations at high current [41,42]. The 360 and 962 cm^−1^ peaks disappeared, and this may be due to the originally weak Raman band signals being blocked by the various oxide formations. The anode showed a greater reduction, as clearly seen in Appendix A.

The XPS spectra of the anode after the stability test are shown in Figure 7c–c-3. In the Co 2p spectrum in Figure 7c-1, the Co 2p3/2 and 2p1/2 peaks are found at 780.2 and 795.9 eV, and the corresponding satellite peaks are centered at 784.8 and 801.2 eV, indicating the existence of Co(OH)_2_ on the electrode surface after stability tests. The oxide-related peaks were largely increased, as clearly seen in Figure 7c-1. This may be due to the continuous redox reactions in OH^−^-containing solution [84]. In the Fe spectrum, similarly, the proportion of oxidized Fe species was also largely increased and taller than the Fe 2P peaks, as seen in Figure 7c-2. In the B 1s and P 2p spectra, increased B and P oxidation peaks were similarly observed, as seen in Figure 7c-3,c-4, resulting from the intense oxidation during the water-splitting process. For the O 1s spectrum, as seen in Appendix A, the increased portion of M-O bonds further confirmed the formation of oxidation states [85]. The XRD pattern after the stability test is provided in Figure 7d, which can be compared with that in Figure 3b before the stability test. Generally, the intensity of XRD diffraction peaks was reduced, and many diffraction peaks disappeared. For example, the diffraction peaks at 22.8, 33.7, 38.4, 41.0, and 42.5° were not visible after the stability test due to the reconstruction of the surface by the formation of an amorphous oxide layer during the water-splitting operation [37]. The peak intensity at 26.7, 27.6, 29.5, and 35.1° decreased, again indicating the formation of an amorphous oxide layer. The oxidation of CoFeBP MF surfaces can generally occur in the initial stages of WS operations, as discussed previously, with the oxidation peak formation, and it stabilizes after multiple operations. Indeed, surface oxidation improves the OER performance as the oxidated species can act as OER active sites in the catalytic process [37,86]. The LSV performance of the CoFeBP electrode after stability testing was examined in 1.0 M KOH, as seen in Appendix A, and only minor differences were observed. 

## 4. Conclusions

In summary, a CoFeBP micro flower (MF) was successfully fabricated via hydrothermal synthesis, and the reaction parameters were systematically optimized. The best optimized CoFeBP MF electrode demonstrated comparable 3-E and 2-E performances with the benchmarks of Pt/C and RuO_2_. Specifically, the CoFeBP MF demonstrated an HER overpotential of 20.1 mV at 20 mA/cm^2^ in 1.0 M KOH. The OER overpotential value of the CoFeBP MF electrode was 219 mV at 20 mA/cm^2^ in 1.0 M KOH. While the HER performance was relatively strong, the OER performance indicated room for further improvement. The bifunctional CoFeBP MFs exhibited a cell voltage of 1.60 V at 50 mA/cm^2^ in 1.0 M KOH. The 2-E performance was quite good; it still showed some room for further improvement, as compared with the state-of-the-art electrocatalysts. In addition, the CoFeBP MFs demonstrated excellent durability, stability, and repeatability at high current and high temperatures in high-alkaline solutions. The CoFeBP MFs demonstrated excellent performances in natural seawater and seawater + KOH solutions. The surface oxidation was systematically characterized after the 12 h stability operation at 1000 mA/cm^2^. The improved performance of CoFeBP MFs can also be attributed to the high ECSA by the distinctive micro-flower morphology and improved short-range local crystallinity by the systematic annealing optimization. Synergy between the elements was effectively utilized, i.e., highly active intrinsic properties by electron-enriched Co and Fe active sites with the incorporation of B and P. 

## Figures and Tables

**Figure 1 nanomaterials-14-00698-f001:**
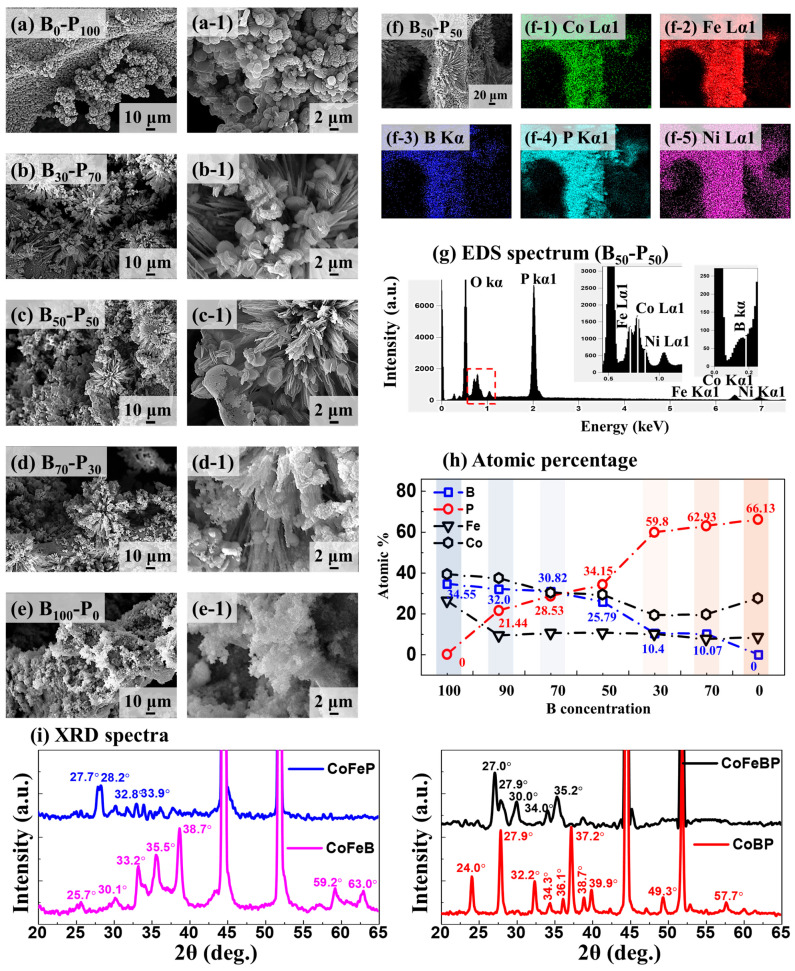
CoFeBP electrode fabrication with B-P concentration variation. Total molarity of B and P was fixed at 12 mM: B_50_-P_50_ indicates 6 mM B and P, namely CoFeB_50_P_50_. Other parameter variations can be found in the Appendix A. (**a**–**e**) SEM images of CoFeBP electrodes as labeled. (**a-1**–**e-1**) Corresponding enlarged SEM images. (**f**–**f-5**) SEM image of B_50_-P_50_ and corresponding EDS phase maps of Co Lα1, Fe Lα1, B Kα, P Kα1, and Ni Lα1. (**g**) EDS spectrum of CoFeB_50_P_50_. (**h**) Atomic percentage summary plots of Co, Fe, B, and P. (**i**) XRD spectra of CoFeBP, CoBP, CoFeP, and CoFeB electrodes.

**Figure 2 nanomaterials-14-00698-f002:**
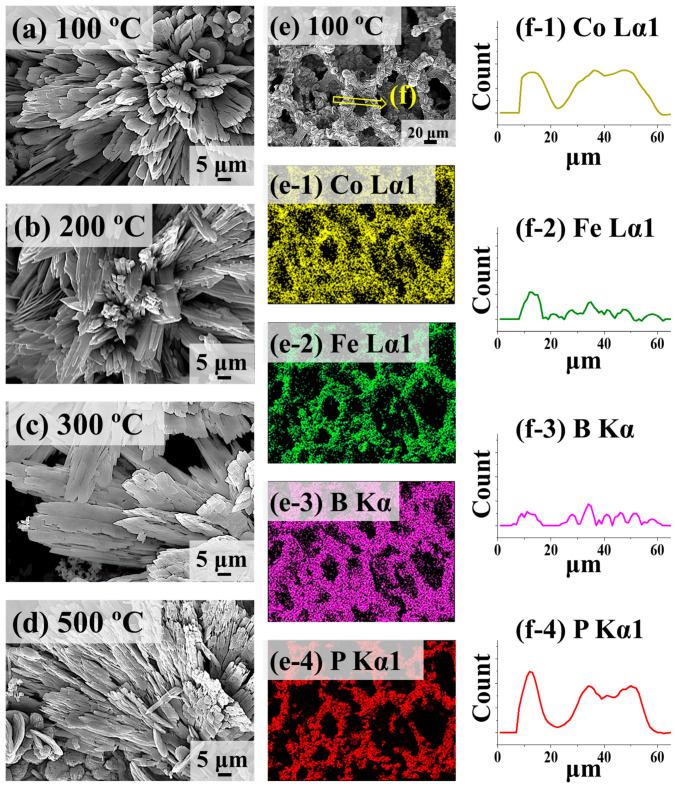
CoFeB_50_P_50_ micro-flower (MF) electrodes with annealing temperature variation. (**a**–**d**) SEM images of CoFeB_50_P_50_ MFs after annealing at different temperatures as labeled. (**e**–**e-4**) SEM image and corresponding EDS phase maps of 100 °C annealed sample. (**f-1**–**f-4**) EDS elemental line profiles from the location as indicated in (**e**).

**Figure 3 nanomaterials-14-00698-f003:**
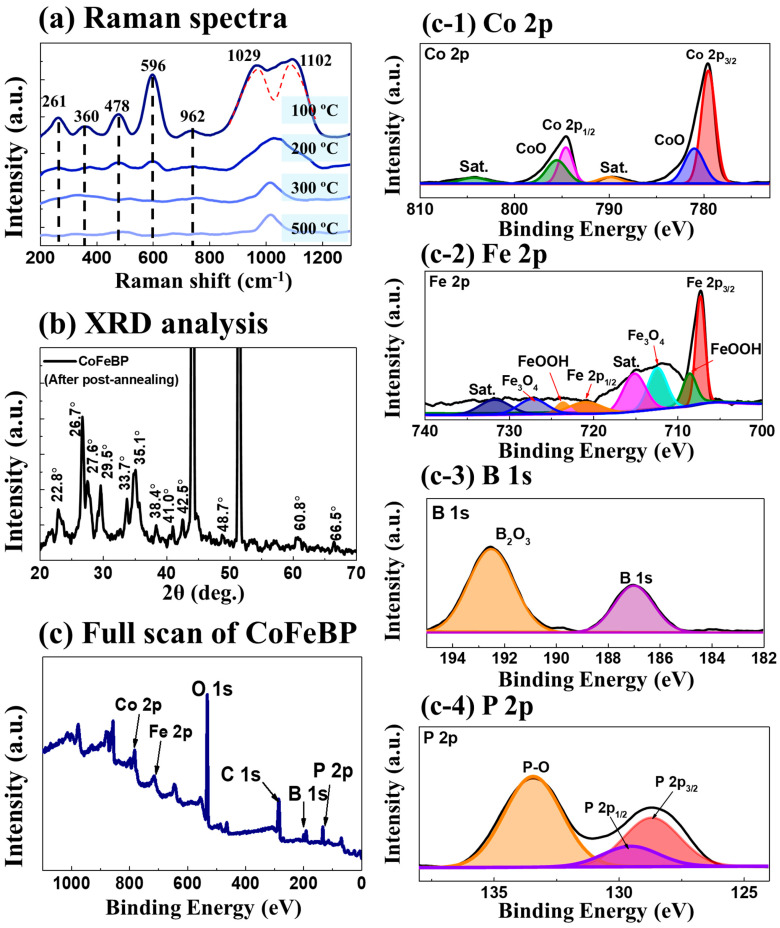
Surface characterizations for the best CoFeBP electrode annealed at 100 °C. (**a**) Raman spectra. (**b**) XRD spectrum. (**c**–**c-4**) XPS spectra of 100 °C CoFeB_50_P_50_ MF.

**Figure 4 nanomaterials-14-00698-f004:**
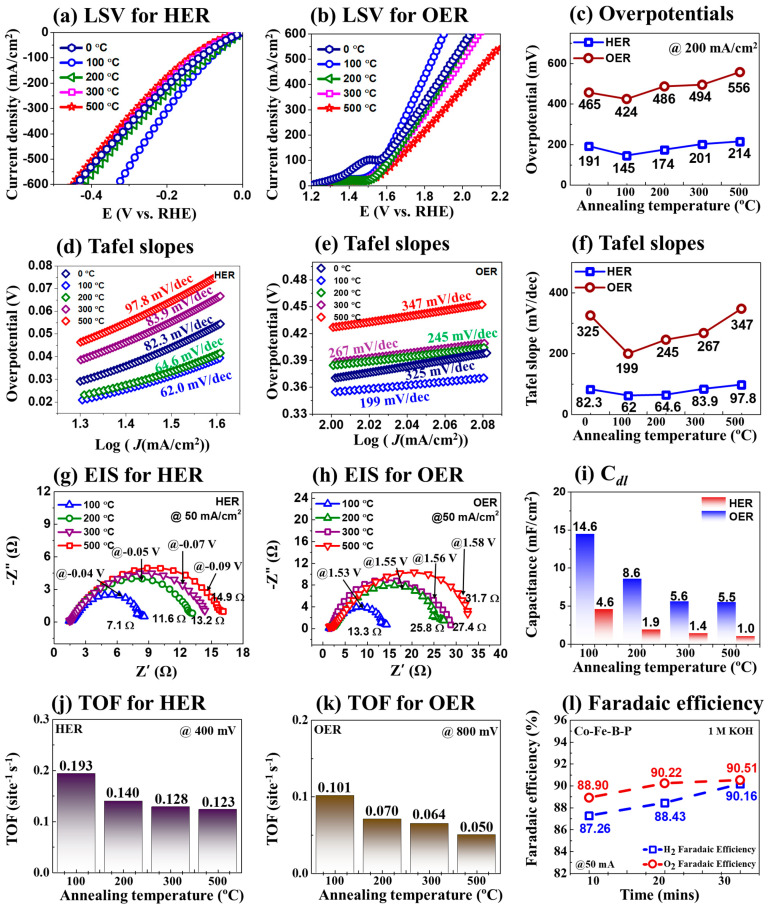
The 3-E electrochemical characterizations of the CoFeB_50_P_50_ MF electrodes in 1 M KOH. (**a**,**b**) HER and OER LSV curves in the post-annealing temperature variation set. (**c**) HER and OER LSV value summary at 200 mA/cm^2^. (**d**,**e**) HER and OER Tafel slope plots. (**f**) Corresponding Tafel slope values. (**g**,**h**) EIS plots. (**i**) Double-layer capacitance (C*_dl_*) obtained from the corresponding CV curves. (**j**,**k**) HER and OER turnover frequency (TOF) plots. (**l**) HER and OER Faradaic efficiency (FE).

**Figure 5 nanomaterials-14-00698-f005:**
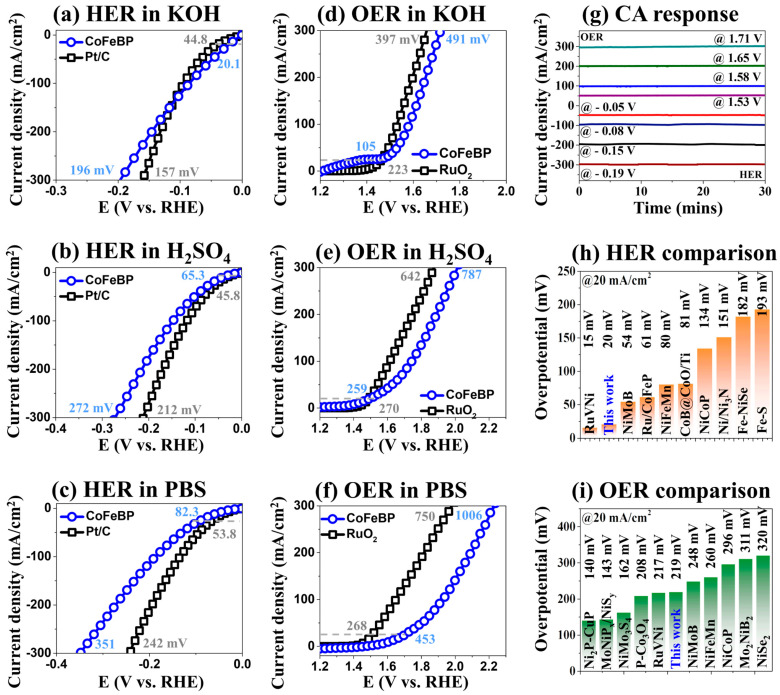
The 3-E electrochemical performance of best CoFeB_50_P_50_ MF electrodes of 100 °C annealed in different media. (**a**–**c**) HER activities in 1 M KOH, 0.5 M H_2_SO_4_ and 1 M PBS electrolyte. (**d**–**f**) 3-E OER activities. (**g**) HER/OER CA responses in 1 M KOH. (**h**,**i**) 3-E HER/OER performance comparisons with the state-of-the-art electrodes at 20 mA/cm^2^.

**Figure 6 nanomaterials-14-00698-f006:**
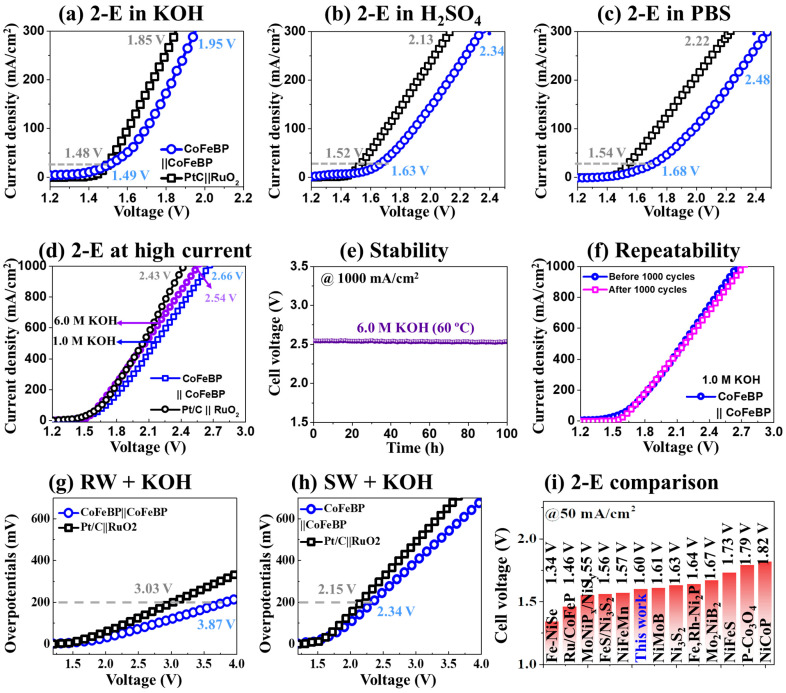
(**a**–**c**) 2-E activity in different media. (**d**) LSV up to 1000 mA/cm^2^. (**e**) Stability test in 6.0 M KOH at 60 °C for 100 h. (**f**) 2-E repeatability in 1.0 M KOH after 1000 cycles. (**g**) 2-E catalytic performance in natural seawater (SW) and river water (RW). (**h**) 2-E performance in SW with the addition of 1 M KOH. (**i**) 2-E performance comparison with the state-of-the-art electrodes at 50 mA/cm^2^.

**Figure 7 nanomaterials-14-00698-f007:**
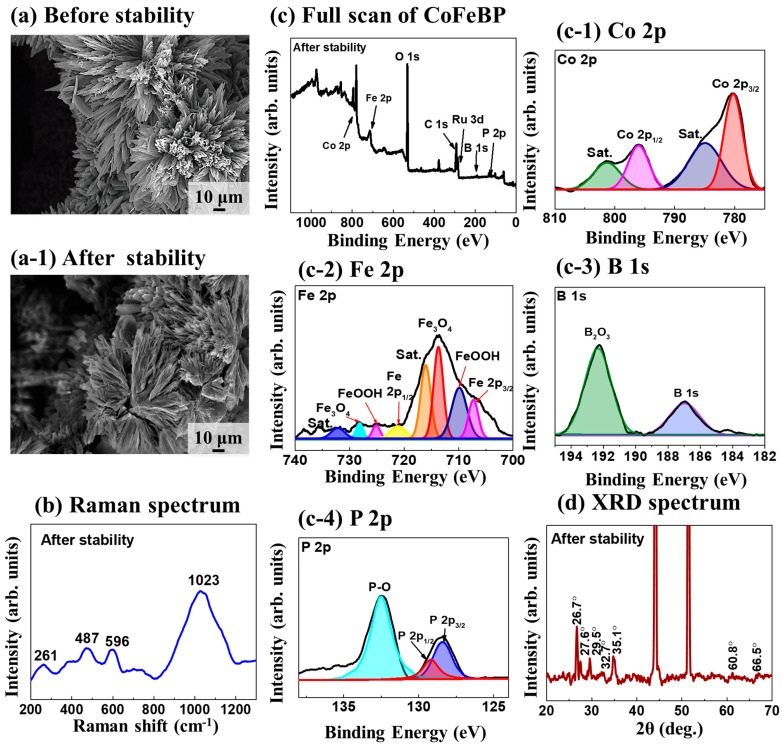
Characterizations of CoFeBP MF after the stability for 12 h at 1000 mA/cm^2^. (**a**,**a-1**) SEM images of CoFeBP (anode) before and after stability test. (**b**) Raman spectrum of anode after stability test. (**c**–**c-4**) XPS spectra of Co 2p, Fe 2p, B 1s and P 2p. (**d**) XRD pattern of anode after stability test.

**Table 1 nanomaterials-14-00698-t001:** Summary of 3-E HER/OER electrochemical properties of the best CoFeBP electrode in 1 M KOH solution.

Electrochemical Properties	HER	OER
EIS values	7.1 Ω	13.3 Ω
Tafel slopes	62 mV/dec	199 mV/dec
C*_dl_* values	4.6 mF/cm^2^	14.6 mF/cm^2^
ECSA	28.75 cm^2^	91.25 cm^2^
TOF values (at 500 and 800 mA/cm^2^)	0.193 site^−1^s^−1^	0.101 site^−1^s^−1^
Faradic efficiency (FE)	90.51%	90.16%

**Table 2 nanomaterials-14-00698-t002:** The 3-E hydrogen evolution reaction (HER) electrocatalytic performance comparison with various transition metal-based catalysts in 1.0 M KOH.

Electrocatalyst	Overpotentials [mV]			Year	References
	@20 mA/cm^2^	@50 mA/cm^2^	@200 mA/cm^2^		
RuVNi	15	26	48	2019	[72]
CoFeBP	20.1	46	145	-	This work
NiMoB	54	97	210	2022	[40]
Ru/CoFeP	61	82	-	2020	[19]
NiFeMn	80	121	-	2020	[73]
CoB@CoO/Ti	81	110	181	2017	[74]
NiCoP	134	165	204	2018	[13]
Ni/Ni_3_N	151	232	420	2015	[59]
Fe-NiSe	182	265	-	2022	[75]
Fe-S	193	235	324	2019	[76]

**Table 3 nanomaterials-14-00698-t003:** The 3-E oxygen evolution reaction (OER) electrocatalytic performance comparison with various transition metal-based catalysts in 1.0 M KOH.

Electrocatalyst	Overpotentials [mV]			Year	References
	@20 mA/cm^2^	@50 mA/cm^2^	@200 mA/cm^2^		
Ni_2_P-CuP	140	190	-	2021	[3]
MoNiP_x_/NiS_y_	143	156	221	2021	[77]
NiMo_3_S_4_	162	252	617	2022	[78]
P-Co_3_O_4_	208	295	330	2018	[18]
RuVNi	217	227	312	2019	[72]
CoFeBP	219	303	426	-	This work
NiMoB	248	267	500	2022	[40]
NiFeMn	260	291	352	2020	[73]
NiCoP	296	328	370	2018	[13]
Mo_2_NiB_2_	311	342	-	2021	[2]
NiSe_2_	320	521	-	2022	[75]

**Table 4 nanomaterials-14-00698-t004:** The 2-E (CoFeBP‖CoFeBP) cell voltage summary of LSV performance of the best CoFeBP electrode in different electrolytes.

Electrolytes	@200 mA/cm^2^	@1000 mA/cm^2^
1 M KOH	1.84 V	2.66 V
6 M KOH	1.76 V	2.54 V
0.5 M H_2_SO_4_	2.13 V	-
1 M PBS	2.25 V	-
River water	-	-
River water + 1 M KOH	3.87 V	-
Seawater	-	-
Seawater + 1 M KOH	2.34 V	-

**Table 5 nanomaterials-14-00698-t005:** The 2-E electrocatalytic performance comparison of overall water splitting with various transition metal-based catalysts in 1.0 M KOH.

Electrocatalyst	Electrolyte	Cell Voltage (V)		Year	References
		@50 mA/cm^2^	@200 mA/cm^2^		
Fe-NiSe	1 M KOH	1.34 V	1.61 V	2022	[75]
Ru/CoFeP	1 M KOH	1.46 V	-	2020	[19]
MoNiP_x_/NiS_y_	1 M KOH	1.55 V	1.88 V	2021	[77]
FeS/Ni_3_S_2_	1 M KOH	1.56 V	-	2022	[80]
NiFeMn	1 M KOH	1.57 V	1.66 V	2020	[73]
CoFeBP	1 M KOH	1.60 V	1.84 V	-	This work
NiMoB	1 M KOH	1.61 V	1.96 V	2022	[40]
Ni_3_S_2_	1 M KOH	1.63 V	-	2018	[81]
Fe,Rh-Ni_2_P	1 M KOH	1.64 V	1.85 V	2022	[82]
Mo_2_NiB_2_	1 M KOH	1.67 V	-	2021	[2]
NiFeS	1 M KOH	1.73 V	-	2022	[83]
P-Co_3_O_4_	1 M KOH	1.79 V	-	2018	[18]
NiCoP	1 M KOH	1.82 V	-	2018	[13]

## Data Availability

Data will be available upon request.

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
