# Peer review of "CoFeBP Micro Flowers (MFs) for Highly Efficient Hydrogen Evolution Reaction and Oxygen Evolution Reaction Electrocatalysts"

_nanomaterials, 2024, doi:10.3390/nano14080698_

Round 1

Reviewer 1 Report (New Reviewer)

Comments and Suggestions for Authors

In this work, the authors prepare CoFeBP micro flowers for efficient water splitting. Also, a two-electrode water splitting is conducted. Some issues should be well solved before acceptance.

1.     The units like mA/cm2 should be mA cm-2. Mistakes about superscripts and subscripts should be corrected. Similar mistakes should be corrected through the manuscript.

2.     To highlight the importance of green hydrogen and improve the readability of the introduction, the authors are suggested to combine some data about global hydrogen requirement and economics in the background, which can refer to this work 10.1039/D3EE02695G.

3.     The oxygen species of materials are critical for the performance of water splitting. The authors should fit O 1s XPS spectra before and after OER to demonstrate the underlying mechanism.

4.     The analysis about EIS is too simple. The EIS can well reflect the electronic conductivity of the materials. Please refer to 10.1002/cey2.465 for more discussion.

Comments on the Quality of English Language

Minor editing of English language required

Author Response

Manuscript ID: nanomaterials-2952624

Dear reviewer #1

            Thank you for your time and efforts to review our manuscript, entitled “CoFeBP micro flowers (MFs) for highly efficient hydrogen evolution reaction and oxygen evolution reaction electrocatalysts”. We sincerely appreciate your constructive comments and have incorporated them into the revised version. Especially, we have enriched the introduction discussion and expanded experimental work as suggested. We certainly consider other comments as well. The detailed answer and action taken are provided below.

            Once again, thank you for your patience and kindness throughout the review process.

Sincerely,

Jihoon Lee , Prof. Ph.D.

Professor

Department of Electronic Engineering,

Kwangwoon University, South Korea

Reviewer 2 Report (New Reviewer)

Comments and Suggestions for Authors

In this work, S. Lin et al. successfully constructed CoFeBP micro flower (MF) via a hydrothermal synthesis optimizing the reaction parameters to obtain a catalyst exhibiting very high HER and OER performances, as demonstrated by the overpotential values obtained at 20 mA cm-2 (219mV for OER and 20mV for HER).  The authors investigate also a 2-electrode system, which shows a cell value of 1.6V at 50 mA cm-2. The manuscript has been well organized and the conclusions are supported by the experimental data. I recommend the publication of the present manuscript after addressing the following minor revisions:

1. The authors affirm in the introduction (page 2, line 60), that ‘the incorporation of P with the strong electronegativity can be largely beneficial for the proton-coupled electron transfer for HER’. Has the incorporation of P an effect also towards OER? Could the P incorporation induce through its strong electronegativity a partially positive charge (d+) in the transition metal which could improve the adsorption of oxygen intermediates?

2. The overpotential reported by authors is taken at 20 mA cm-2, however most of literature compares the overpotential at 10 mA cm-2. Why authors choose this current density value and 50 mA cm-2 for the two-electrode system?

3. How did the authors calculate the TOF values? Did they consider all the total mass of catalyst as active (TOF total metal), or only the amount of electroactive sites in the calculation? This detail should be provided in the article.

4. Some flaws are present in the text, for example on page 2 line 87 ’on Ni form’ should be corrected (Ni foam), on page 5 line 180  ‘The annealing duration was 30 min based on the annealing duration optimization’ is a bit redundant (annealing duration repeated), on page 7 line 227 ‘In the Co 2p spectrum in Figure 3(c-1), the binding energy (BE) of Co 2p1/2 and Co 2p3/2 was found at 793.6 and 778.5 eV as compared with the pristine peak obtained in XPS handbook (793.3 and 778.3 eV) [41], exhibiting positive shifts of 0.3 and 0.3 eV’. Last shift value is not different of  ‘0.3 eV with the pristine value of 778.3 eV, but of 0.2 eV, please correct.

Please revise all the text.

5. On pag.7, line 259, authors affirm that ‘the 100 °C CoFeBP MFs showed the lowest HER/OER Tafel values of 62.0 and 199 mV/dec’. While the HER Tafel value is quite good, the OER value is quite high in my opinion. Do the authors think that this could be a negative point for the catalyst?

6. Why the authors decided to not apply the iR correction in LSV studies? This should improve the performance, as authors affirm in the supporting information. 

7. Authors used Ag/AgCl as reference electrode, Hg/HgO is more stable in KOH media, then it should be preferrable.

8. In Table 3 reporting the comparison of the OER electrochemical performance the catalyst obtained in this study (CoFeBP) shows higher overpotential when compared with the majority of others reported. Is there room for further improvement? If yes, how?

9.  In the conclusions (page 15, line 505) authors affirm that ‘The 2-E cell voltage of 1.60 V at 50 mA/cm2 was observed in 1.0 M KOH, which ranks the CoFeBP MF as a good transition metal-based electrode’.

The authors affirm that the CoFeBP MF electrode is ‘good’. Do the authors envision some improvements (structural/compositional/morphological ecc.) to further improve the performance?

Also, in view of a possible  large scale application, how much should this potential value be improved?

10. The authors are advised to cite in the introduction recent published relevant articles focused on transition metal based electrocatalysts for water electrolysis, such as Catalysis Today 2023, 423, 113929 (https://doi.org/10.1016/j.cattod.2022.10.011), Chemical Engineering Journal 2023, 462, 142177 (https://doi.org/10.1016/j.cej.2023.142177) and others.

Author Response

REVIEW REPLY:

Manuscript ID: nanomaterials-2952624

Dear reviewer #2

            Thank you for your time and efforts in reviewing our manuscript. We are grateful for the opportunity to benefit from your knowledge and insight, which is critical in improving the quality of our research work. We certainly consider all the comments and have made necessary changes as suggested. The details are provided below.

Once again, thank you for your diligent review and kindness during the review process.

Sincerely,

Jihoon Lee , Prof. Ph.D.

Professor

Department of Electronic Engineering,

Kwangwoon University, South Korea

Reviewer 3 Report (New Reviewer)

Comments and Suggestions for Authors

In this manuscript, the authors reported the hydrothermal synthesis of CoFeBP micro flowers (MFs) and their application as electrocatalysts for efficient hydrogen evolution reaction (HER) and oxygen evolution reaction (OER). Noteworthy, the various hydrothermal synthesis and post-annealing parameters were thorough investigated to achieve the best-performing CoFeBP electrocatalyst towrd HER and OER. The optimised catalyst was further appraised under near-industrial conditions, showing promise for practical use. Overall, this work has good novelty and is worthy to be published in Nanomaterials. Before possible acceptance, the below detailed comments need to be addressed to further improve the quality of the manuscript.

1. This reviewer is interested in the crystalline phase structure of the materials reported. While discussion was made on this point, with results supported mainly by XRD, the exact phase (alloy, boride, phosphide, or other phase?) of the as-prepared CoFeBP was not clearly elaborated.

2. To appeal to a broader readership, recent works on water electrolysis can be referenced in Introduction (e.g., InfoMat, 2023, DOI: 10.1002/inf2.12494).

3. Figure 3, some of the XPS analysis might require revision. For instance, Figure 3c-1 and 3c-2, the fitted sub-peaks seem to not add up to the sum of the total peak. This is especially for the analysis of the Co 2p XPS data, where some additional sub-peaks might be missing from the current analysis. Please also check other XPS analysis of the manuscript for similar issues.

4. The study under near-industry conditions (60 oC and 6M KOH) is really interesting. How were these tests performed? More experimental details should be provided. Related works on water electrolysis at elevated temperatures and highly alkaline solutions could be referenced (e.g., ChemSusChem, 2024, DOI: 10.1002/cssc.202301534).

5. The authors mentioned “CO2 reduction reaction (CO2RR) [12,13]” in Introduction (line 57), which, however, is not relevant to the main focus of research in this manuscript (HER/OER) and hence is suggested to be revised (or deleted).

6. Figure 4d, for the y axis, oftentimes, the overpotential should be presented as positive values. Please also check other HER data for similar issues.

Comments on the Quality of English Language

Minor revision to the text is required. For instance, line 12, “water electrocatalyst” might need to be revised into “water-splitting electrocatalyst” for better accuracy.

Author Response

REVIEW REPLY:

Manuscript ID: nanomaterials-2952624

Dear reviewer #3

            Thank you for your insightful and valuable feedback on reviewing our manuscript. We are honored to receive such constructive comments and have made necessary changes as suggested. Specifically, we have corrected the errors which appeared in the experimental results and also improved the clarity of the introduction section. We certainly considered other suggestions as well. The details can be found below.

Once again, we sincerely appreciate your kindness and patience throughout the review process.

Sincerely,

Jihoon Lee , Prof. Ph.D.

Professor

Department of Electronic Engineering,

Kwangwoon University, South Korea

This manuscript is a resubmission of an earlier submission. The following is a list of the peer review reports and author responses from that submission.

Round 1

Reviewer 1 Report

Comments and Suggestions for Authors

This work reports the preparation of CoFeBP MFs on a porous nickel substrate by hydrothermal and annealing processes as an electrocatalyst for the OER and HER, which have excellent electrocatalytic performance under industrial conditions. However, some tips should be considered before it is published to make this paper more systematic and reasonable.

1. In this experiment, the porous nickel foam is used as the substrate, whether Ni is involved in the reaction in the preparation of the catalyst, the naming of Co-Fe-B-P should be reconsidered, and the physical characterization of Ni should be supplemented to analyze the role of Ni in the catalyst.

2. The authors should indicate the PDF card corresponding to the XRD diffraction peaks in the text or the illustrations, and the XPS section of the P fitting is wrong and should be re-analyzed.

3. In the Raman analysis section, the author mentions “The CoFeBP is a new material, and thus a Raman of CoFeBP was not referenced in literature” which is incorrect, and it is recommended that the authors consult the relevant literature and re-analyze the Raman spectroscopy.

4. Regarding the electrochemistry part, what is the significance of the test of acidic and neutral pH? In Fig 4.g the 30-minute stability test does not indicate that the catalyst has excellent stability.

5. In the conclusion, the author indicates the ranking of the catalyst material among the best catalysts to date, whether this ranking is well-founded, and if not, suggests a change in the expression.

Comments on the Quality of English Language

Author Response

Manuscript ID: nanomaterials-2802971

Dear reviewer #1

Thank you for your time and efforts to review our manuscript entitled “CoFeBP micro flowers (MFs) for highly efficient hydrogen evolution reaction and oxygen evolution reaction electrocatalysts”. Your constructive and positive feedback can certainly improve the quality of our research work. We have adapted all the comments and have made necessary changes which can be found below.

We sincerely appreciate your time and dedication throughout the review process.

Sincerely,

Jihoon Lee , Prof. Ph.D.

Professor

Department of Electronic Engineering,

Kwangwoon University, South Korea

http://aqnmol.kw.ac.kr/index.html

Email: jihoonlee@kw.ac.kr

Tel: 82-2-940-5297

Fax: 82-2-942-5235

Reviewer 2 Report

Comments and Suggestions for Authors

In this study, Lin et al. synthesized CoFeBP microflowers (MFs) as catalysts for the hydrogen evolution reaction (HER) and oxygen evolution reaction (OER). They observed low overpotentials for HER/OER and high stability under harsh operational conditions, attributing the enhanced catalytic performance of CoFeBP MFs to their flower-like morphology and good crystallinity. However, before publication, several major issues need to be addressed:

1.    The chemical composition and structure of the CoFeBP MFs sample should be clarified, as strong C1s and O1s peaks are evident in Figure 2i.

2.    The authors should ensure a correct analysis of the XPS data in Figure 2, as the assignment of peaks and the deconvolution of spectra are completely wrong. For example, the Co2p3/2 peak at 779.5 eV in Figure 2i-1 should be assigned to Co3O4 rather than metallic Co with charge transfer. Further, the deconvolution of Fe 2p spectra in Figure 2i-2 is completely wrong (10.1016/j.rinp.2019.102498). In Figure 2i-4, the peak area ratio of P2p1/2 and P2p3/2 should be 1:2.

3.    The significant disparity in Cdl values for the best CoFeBP electrode for HER and OER (Table S1) should be explained, including any potential changes in electrode morphology under different reaction conditions.

4.    It is important to investigate whether there was any dissolution of the CoFeBP MFs material during stability testing.

5.    The suitability of this catalyst as an anode catalyst for CO2 reduction can be discussed to enhance the manuscript's appeal to readers (10.1016/j.xcrp.2022.100949; 10.1021/acscatal.2c03842).

Comments on the Quality of English Language

Moderate editing of English language is required.

Author Response

Manuscript ID: nanomaterials-2802971

Dear reviewer #2

Thank you for your diligent review of our manuscript entitled “CoFeBP micro flowers (MFs) for highly efficient hydrogen evolution reaction and oxygen evolution reaction electrocatalysts”. We are grateful for your insightful comments, which undoubtedly enhance the overall quality of the manuscript. We have carefully considered all the suggestions and have made the necessary changes. The answer and action taken are provided below.

We sincerely appreciate your time and patience during the review process.

Sincerely,

Jihoon Lee , Prof. Ph.D.

Professor

Department of Electronic Engineering,

Kwangwoon University, South Korea

http://aqnmol.kw.ac.kr/index.html

Email: jihoonlee@kw.ac.kr

Tel: 82-2-940-5297

Fax: 82-2-942-5235

Reviewer 3 Report

Comments and Suggestions for Authors

The authors have describe some interesting results for water electrolysis with the CoFeBP-MF catalysts that they have prepared. The results cover a wide range of experimental techniques, all with useful information. My only fundamental issue is the need for an estimate of the catalytic activity on a true area basis, i.e., electrochemically active surface area, together with a comparison of the values with those of other researchers. As noted below, the turnover frequency, as calculated in this work, is not useful in this regard.

Specific comments

Page 1, Introduction

Ru-based electrodes are the standard benchmark 38 catalysts but their high costs with rare earth stock ...

Please change "rare earth stock" to "low earth abundance."

The use of the word "charming" should be reconsidered. Perhaps "desirable," "attractive," "fascinating," or "promising."

The use of "2-E" and "3-E" is non-standard and obscure. Please explain these. It is finally explained on page 7.

Page 2

"The improved performance can be linked to the high ECSA offered by the micro flower (MF) morphology and synergy between the elements utilized."

It appears that an attempt has been made to quantify the electrocatalytic activity on a true area basis, i.e., electrochemically active surface area (ECSA), and values have been given. However, I was unable to discover how the latter were calculated. This is a crucial point that needs to be addressed in detail. Were they calculated from the double-layer charging currents or capacitances? If so, what was the conversion factor, and how was this chosen? The activities normalized to ECSA need to be compared with values from other researchers.

Page 4

"Among the 2θ range from 20 to 65 o, the appearance of multiple peaks in all 156 samples might indicate the fabrication of amorphous polycrystalline structures [33]."

The phrase "amorphou polycrystalline structures" is contradictory and meaningless.

Page 8

It is not clear what HER and OER Cdl values are. Please explain in more detail.

Page 9

The turnover frequencies are somewhat meaningless, because they were calculated on the basis of the total number of transition metal atoms in the catalyst. Most of these are not available for the reaction. It is necessary to estimate the actual number of available surface sites. It is also necessary to compare the TOF values with values from other work.

Page 13

"After the continuous 100-hour 406 operation, small particles were observed at the bottom of cell."

Were the particles analyzed?

Comments on the Quality of English Language

The English is mostly understandable, but there are a number of minor spelling and grammatical errors, which need to be corrected.

Author Response

Please find the answers in the attached file.

Round 2

Reviewer 2 Report

Comments and Suggestions for Authors

This manuscript can be accepted.

Author Response

Thank you for the positive comments.

Reviewer 3 Report

Comments and Suggestions for Authors

The authors have responded adequately to my comments.